# Gene Analysis of *Listeria monocytogenes* Suspended Aggregates Induced by *Ralstonia insidiosa* Cell-Free Supernatants under Nutrient-Poor Environments

**DOI:** 10.3390/microorganisms9122591

**Published:** 2021-12-15

**Authors:** Qun Li, Ailing Guo, Yi Ma, Ling Liu, Wukang Liu, Yuan Zhong, Yawen Zhang

**Affiliations:** 1National Research and Development Center for Egg Processing, College of Food Science and Technology, Huazhong Agriculture University, Wuhan 430070, China; liqun0617@126.com (Q.L.); 18438605237@163.com (L.L.); liu_wukang@126.com (W.L.); zhongyuan2329@126.com (Y.Z.); zywwwww1998@126.com (Y.Z.); 2Hubei Provincial Institute for Food Supervision and Test, Wuhan 430070, China

**Keywords:** *Listeria monocytogenes*, *Ralstonia insidiosa* cell-free supernatants cultured in 10% TSB medium, suspended aggregates, RNA-seq, formation mechanism, *cheAY*

## Abstract

*Listeria monocytogenes* is a zoonotic food-borne pathogen. The production of food-borne pathogenic bacteria aggregates is considered to be a way to improve their resistance and persistence in the food chain. *Ralstonia insidiosa* has been shown to induce *L. monocytogenes* to form suspended aggregates, but induction mechanisms remain unclear. In the study, the effect of *R. insidiosa* cell-free supernatants cultured in 10% TSB medium (10% RIS) on the formation of *L. monocytogenes* suspended aggregates was evaluated. Next, the Illumina RNA sequencing was used to compare the transcriptional profiles of *L. monocytogenes* in 10% TSB medium with and without 10% RIS to identify differentially expressed genes (DEGs). The result of functional annotation analysis of DEGs indicated that these genes mainly participate in two component system, bacterial chemotaxis and flagellar assembly. Then the reaction network of *L. monocytogenes* suspended aggregates with the presence of 10% RIS was summarized. The gene-deletion strain of *L. monocytogenes* was constructed by homologous recombination. The result showed that *cheA* and *cheY* are key genes in the formation of suspended aggregates. This research is the preliminary verification of suspended aggregates’ RNA sequencing and is helpful to analyze the aggregation mechanisms of food-borne pathogenic bacteria from a new perspective.

## 1. Introduction

*Listeria monocytogenes* is a food-borne pathogen that is liable for a disease called listeriosis, a rare but severe disease in humans, who can become infected by ingesting contaminated food products. As a motile Gram-positive rod, *L. monocytogenes* can grow at a wide range of temperatures (1 to 45 °C) and survives extreme growth conditions comparatively well, from pH 4.5 to 9 and into 10% NaCl, but optimum growth occurs at 0.5% NaCl and neutral pH [1]. Based on the characteristics of tolerance to high salt, low temperature and poor nutrition, *L. monocytogenes* is present widely in the environment, as it can be isolated from water, soil, silage, vegetables and food-processing environments [2]. In Denmark in 2016, two confirmed incidents of foodborne infection with *L. monocytogenes* were reported [3]. In 2019, an incident with contaminated chilled meat caused 1060 ill-nesses and 216 deaths [4]. According to the data on the number of biohazards in foods reported or recalled globally from 2008 to 2018, the number of hazards of *L. monocytogenes* surpassed *Salmonella* and *Escherichia coli* and ranked first, with a total of 387 cases [5]. Therefore, the presence of *L. monocytogenes* in food-processing environments is a major burden on the food industry and a challenge to food safety.

*Ralstonia insidiosa* was first isolated by Coenye et al. in the United States in 2003; it belongs to non-fermenting, Gram-negative and aerobic bacteria and occasionally causes infection as an opportunistic pathogen [6]. *R. insidiosa* inhabits a broad range of the water environment, such as industrial pure water, laboratory pure water, river, pond and soil [7]. It has been shown that *R. insidiosa* is a link for the biofilm formation of *L. monocytogenes*, *Salmonella enterica* and *E. coli* [8]. Currently, there are few studies and reports on *R. insidiosa*.

Microorganisms are not insular entities, but they usually grow together in multi-species biofilm communities with complex classification [9]. The comparison between aggregation (i.e., adhesion in aquatic environments without a surface) and biofilm formation (i.e., adhesion to the surface) is presently unclear. Moreover, they may be related to the surface where microorganisms live, and aggregation is one of the processes of biofilm formation [10]. In the natural liquid environment, bacterial cells usually combine with the wet surface and interface in the shape of multicellular aggregates, some of which are called suspended aggregates [11]. Suspended aggregates are regarded as non-classical biofilms, in which microbial cells strongly express extracellular polymeric substances (EPS) components but do not directly contact the surface [12,13]. Some bacteria can form flocs through the aggregation process with the help of other bacteria [14]. The non-aggregating bacterium *Xenophilus* was also discovered to aggregate, but only with the assistance of *Methylobacterium* [10]. *Methylobacterium* may act as a bridging microorganism in cross species aggregate formation. In 2018, Trunk et al. found that the auto-aggregation or adhesion of the same bacteria to each other not only depends on environmental conditions, but also on different bacterial species [15]. However, *L. monocytogenes* can be induced to form suspended aggregates by *R. insidiosa* but cannot form suspended aggregates by itself [16]. In these well-organized structures, microorganisms may be free from environmental pressure, such as disinfectants and cleaning, and ecological interactions may occur among different species [17]. It is currently reported that suspended aggregates are the root cause of chronic diseases [18,19], and suspended aggregates have attracted more and more attention.

The production of suspended aggregates may be caused by the production of secondary metabolites by these bacteria to promote the formation of suspended aggregates [20]. Based on the research of Guo et al. [16], we evaluated the effect of *R. insidiosa* cell-free supernatants cultured in 10% TSB medium (10% RIS) on the formation of suspended aggregates formed by *L. monocytogenes*. RNA sequencing (RNA-seq) can be applied to determine the transcriptional changes in the interactions between pathogenic bacteria and microbial interspecies [21]. Then RNA-seq was utilized for preliminarily analyzing the impact of the formation of *L*. *monocytogenes* suspended aggregates induced by 10% RIS on the transcriptome of *L*. *monocytogenes*. This study can explain the factors affecting the formation of suspended aggregates by *L. monocytogenes* in terms of the genetic level, and bring new ideas for the prevention and control of food-borne pathogens in food processing environments. Investigating *L. monocytogenes* aggregation may provide insight into how the bacterium successfully transitions between its different niches, as well as reveal general mechanisms underlying non-surface-dependent multicellular community formation.

## 2. Materials and Methods

### 2.1. Bacterial Strains and Growth Media

The *L*. *monocytogenes* isolate used, with designated strain name “100”, was previously isolated from a milk sample. Previous experiments in our laboratory identified this as a serotype 1/2a strain with strong biofilm and suspended aggregate formation ability. *R. insidiosa* ATCC 49129 was purchased from the American Type Culture Collection (ATCC). Both strains were stored at −80 °C in tryptic soy broth (TSB; Hope Bio-Technology, Qingdao, China) containing 40% glycerol. Before all experiments, frozen cells were activated and separated on tryptic soy agar plates (TSA; Hope Bio-Technology, Qingdao, China) for 24 h at 37 °C. Then one colony was transferred into TSB overnight at 37 °C, and the cell concentration was adjusted to 10^8^ colony-forming units (CFU)/mL. The concentration of bacteria used in each experiment was 10^8^ CFU/mL, unless otherwise specified. *E*. *coli* DH5α (Takara Bio Inc., Otsu, Shiga, Japan) was used as the standard plasmid host for cloning procedures and was grown in lysogeny broth (LB; Hope Bio-Technology, Qingdao, China) broth and agar. Ampicillin (100 μg/mL) (Biofroxx, Einhausen, Germany) or erythromycin (3 μg/mL) (Solarbio, Beijing, China) were added to agar media or broth as required.

### 2.2. Preparation of Suspended Aggregates

#### 2.2.1. Suspended Aggregate Formation of *L*. *Monocytogenes*

As previously described [16], 200 µL of *L. monocytogenes* was cultured into the 90 mm plastic Petri dish containing 20 mL 10% TSB medium. Thereafter, the 90 mm plastic Petri dish was cultured at 28 °C and 60 RPM. The formation of suspended aggregates was checked, compared and photographed daily.

#### 2.2.2. Suspended Aggregate Formation of *L*. *Monocytogenes* Induced by 10% RIS

The *R. insidiosa* colony was transferred into 10% TSB medium for 24 h at 28 °C and 150 RPM. The culture fluid was centrifuged for 15 min at 10,000 RPM. The cell supernatant was filtered with a 0.22 μm cell filter. The supernatant after filtration was cultured by TSA plates and TSB medium to ensure the sterilization effect. A total of 200 µL of the *L. monocytogenes* was cultured into a 90 mm plastic Petri dish with 10 mL 10% RIS and 10 mL 10% TSB medium. Thereafter, the 90 mm Petri dish was cultured at 28 °C with 60 RPM. The formation of suspended aggregates was checked, compared and photographed daily.

### 2.3. Scanning Electron Microscopic Observation of Suspended Aggregates

According to the methods in Section 2.2.1 and Section 2.2.2, the suspended aggregates formed after 24 h were collected. After 5 min of centrifugation at 4000 RPM, the supernatant was removed, and the precipitate was cleaned with a physiological saline solution. The precipitate was fixed with phosphate buffer solutions of 2.5% (vol/vol) glutaraldehyde (pH 7.4, 0.02 M) and placed in a refrigerator at 4 °C. After 12 h, the aggregates were centrifugated at 4000 RPM for 10 min and cleaned with 0.02 M phosphate buffered saline (pH 7.4) for 3 times. Then the obtained precipitate was washed with gradient concentration of 30%, 50%, 70% and 90% (vol/vol) ethanol successively for 10 min each time. The next step was to treat twice with 100% ethanol and remove the ethanol by centrifugation. The bacterial pellet was resuspended twice in 100% tert-butanol for two times and stored up in a refrigerator at 4 °C for 30 min; they were then resuspended in 100% tert-butanol for vacuum freeze-drying for 48 h. the dried powder samples were placed under a scanning electron microscope (JSM-6390LV, JEOL, Tokyo, Japan) for observation.

### 2.4. Preparation of Sequencing Samples

#### 2.4.1. *L. Monocytogenes* Sample

A total of 200 µL of the *L. monocytogenes* was cultured into a 90 mm plastic Petri dish with 20 mL 10% TSB medium. Then the 90 mm plastic Petri dish was cultured at 28 °C and 60 RPM for 24 h. After 20 min of centrifugation at 5000 RPM, the cell supernatant was removed, and then the bacterial pellet was washed twice by physiological saline solution. The sample number was X. Three biological replicate experiments were performed for each group of samples.

#### 2.4.2. Suspended Aggregate Sample

A total of 200 µL of the *L. monocytogenes* was cultured into a 90 mm plastic Petri dish with 10 mL 10% RIS and 10 mL 10% TSB medium. Thereafter, the 90 mm plastic Petri dish was cultured at 28 °C and 60 RPM for 24 h. The suspended aggregates were washed twice by physiological saline solution. The sample number was Y. Three biological replicate experiments were performed for each group of samples.

### 2.5. Extraction and Purification of RNA from Suspended Aggregates or L. monocytogenes

RNA was extracted based on the method described by Zhang et al. [22]. The main reagent used is TRIzol (Invitrogen, Carlsbad, CA, USA), which is a new total RNA extraction reagent that can directly extract total RNA from cells or tissues. All RNAs used for following library preparation were determined to have RNA integrity numbers (RINs) of 6.5 and above.

### 2.6. Library Construction and Sequencing

RNA-seq library preparations were built based on the manufacturer’s manual, and the Ribo-Zero rRNA removal kit (Bacteria, Illumina, San Diego, CA, USA) was applied to the loss of ribosomal RNA. The construction of the RNA-seq library mainly refers to the method of Zhang et al. [22]. The basic process includes RNA interruption, double-stranded cDNA synthesis, terminal repair, dA-tailing, splice, fragment sorting, digestion of dUTP-labeled double strand and PCR amplification. Libraries with different indices were multiplexed and loaded on an Illumina HiSeq equipment based on manufacturer’s manual (Illumina, San Diego, CA, USA). The processing and analysis of sequences were operated by GENEWIZ. According to the procedure of Chen et al. [23], Bowtie2 (v2.2.6) was utilized to index the reference genome sequence of *L. monocytogenes* FSL F6-684 (https://www.ncbi.nlm.nih.gov/nuccore/JOOX00000000 (accessed on 1 December 2019)). All transcriptome raw data were stored at the National Center for Biotechnology Information (NCBI) Sequence Read Archive (SRA) database, registered as PRJNA756162.

### 2.7. Differentially Expressed Genes Analysis

Initially, the known gff annotation file was converted to fasta format. Then the file was used as a reference gene file, and accurate indices were made. The expression level of a gene is directly reflected by its gene abundance. The higher the gene abundance, the higher the gene expression level. Gene expression was calculated by using Htseq software (v0.6.1p1). The software uses FPKM (Expected Number of Fragments Per Kilobase of Transcript Sequence Per Millions base pairs sequenced) to calculate gene expression [24]. For samples with biological duplication, differentially expressed genes analysis was performed by using DESeq2 (v1.6.3) from the Bioconductor software package. Based on Hochberg and Benjamini’s method, the results were screened according to the differential significance criteria (the differential gene expression change was more than 2 times and *p*-value (padj) ≤ 0.05), and the upregulation of the differential expressed genes (DEGs) expression was counted.

### 2.8. GO and KEGG Pathway Enrichment Analyses

Gene Ontology (GO)-enrichment analysis method is GOseq (v1.34.1), which is based on Wallenius non-central hyper-geometric distribution. By estimating the bias of gene length, the probability of GO term enrichment by DEGs can be more accurately calculated. The screening criteria for significant enrichment was padj < 0.05. Then topGO was utilized to plot Directed Acyclic Graph (DAG). Kyoto Encyclopedia of Genes and Genomes (KEGG) is a primary public database of Pathways. Pathway-significance enrichment analysis was conducted by using the KEGG pathway as a unit and using hypergeometric test to find out which pathway was significantly enriched in DEGs as compared with the whole genome background. In the study, scripts in house were utilized to enrich significant DEGs in KEGG pathways.

### 2.9. Construction of Deletion Strains

Genes obtained by transcriptome analysis were used to confirm the function of the gene on the formation of suspended aggregates. The plasmid pMAD used in this study was presented by Gang Wang, the professor of Henan University. Using *L. monocytogenes* 100 DNA as the template, the DNA fragments on both sides of coding DNA sequence encoded *cheA* and *cheY* (*cheAY*) were amplified by PCR. The PCR primers used in this experiment are listed in Appendix A Appendix A. Primers P5F, P5R, P3F and P3R were used for PCR amplification of upstream and downstream homologous arms. Then the 509 bp upstream homologous arm and 401 bp downstream homologous arm were obtained. The two fragments were used as templates for overlapping-extension PCR (SOE-PCR). The SOE-PCR fragments and thermosensitive plasmid pMAD were digested by *EcoR* I and *Bgl* II, respectively [25]. Next, the two fragments of enzyme digestion were ligated and transformed to *E. coli* DH5α and selection on LB agar with ampicillin (100 μg/mL). After purification from *E. coli* positive strains by using Plasmid Mini Kit I D6943 (Omega Bio Tek, Norcross, GA, USA), the resulting plasmids, pMAD-Δ*cheAY*, were electroporated into *L. monocytogenes* competent cells and also in the in-frame Δ*cheAY* mutant with the selection performed on TSA-containing erythromycin (3 μg/mL). Using erythromycin as the screening condition, the *L. monocytogenes* strain containing pMAD-Δ*cheAY* plasmid was subcultured continuously to realize homologous recombination. The isogenic mutant strain with deleted *cheA* and *cheY* gene was named *L. monocytogenes* Δ*cheAY* and confirmed by DNA sequencing on both strands, using primers P5F-P3R and DE-DR, respectively.

### 2.10. The Strain Growth and Motility Assay

For *L*. *monocytogenes* wild-type or mutant strains, growth curves were assayed at 37 °C. Motility assay was carried out as del Campo et al. described, onto a plate containing TSB with 0.3% agar cultured at 28 °C [26]. After 48 h, the bacterial colony diameter was determined with Vernier caliper.

### 2.11. Determination of the Ability to Form Suspended Aggregates

A total of 200 µL of *L*. *monocytogenes* wild-type or mutant strains were cultured into a 90 mm plastic Petri dish with 20 mL 10% TSB medium or with 10 mL 10% RIS and 10 mL 10% TSB medium. Thereafter, the 90 mm plastic Petri dish was cultured at 28 ºC with 60 RPM for 24 h. Then the formation of suspended aggregates was checked, compared and photographed. The aggregation partnerships were studied semi-quantitatively, using aggregation indices [19]. Suspended aggregates were collected in a centrifuge tube, and the OD600 of the solution in the Petri dish was measured. This absorbance value was defined as ODs. Then the remaining solution in the Petri dish was transferred to the centrifuge tube containing suspended aggregates, the bacteria in the centrifuge tube were thoroughly dispersed and mixed by vortex oscillation and the OD600 was measured. The absorbance value was defined as ODt. The aggregation indices were calculated according to the following formula:% aggregation indices = [(ODt − ODs)/ODt] × 100.

### 2.12. Statistical Analysis

Three replicate trials were carried out for each sample, and all samples were repeated three times for the assay of suspended aggregate formation, the growth and motility assay of *L. monocytogenes* wild-type or mutant strains. Except for other explanations in the experiment, the data are generally expressed as mean ± standard error (SE). Student’s *t*-test was utilized to compared the mean value. The *p*-values < 0.05 were identified as a significant difference.

## 3. Results

### 3.1. Preparation and Scanning Electron Microscopic Observation of Suspended Aggregates

Our previous research indicated that *R. insidiosa* induced suspended aggregate formation of *L. monocytogenes* under nutrient-poor environments (10% TSB) at low speed [16]. Then 10% RIS was used instead of *R. insidiosa* to carry out the experiment. *L*. *monocytogenes* did not form suspended aggregates after culturing at low speed (60 RPM) for 72 h (Figure 1). However, in the presence of 10% RIS, *L*. *monocytogenes* formed suspended aggregates after low-speed culture for 24 h (Figure 1). The suspended aggregates in the broth were scattered and small in volume. After 72 h of continuous cultivation, the suspended aggregates gradually disappeared, and the suspended aggregates reached the maximum volume at 24 h of cultivation. This is a new observation and the first demonstration that aggregation can be induced by cell-free supernatants.

The suspended aggregates were fixed with phosphate buffer solutions of 2.5% (vol/vol) glutaraldehyde (pH 7.4, 0.02 M). Scanning electron microscopy was used to observe the change of bacterial surface morphology. The surface of *L. monocytogenes* cells without the role of 10% RIS was almost all smooth and distributed evenly (Figure 2a). After 10% RIS treatment, *L. monocytogenes* aggregated into clusters with EPS (arrow) connections between the cells (Figure 2b).

### 3.2. Quality Control of Sequencing Data

To mine the transcriptome of *L. monocytogenes* growing in suspended aggregates or in floating state, we collected the samples under both conditions after the strain formed suspended aggregates or showed a floating state in plastic Petri dishes. Each sample was collected three times, which was suitable for constructing six cDNA libraries. The Illumina HiSeq platform was then used to perform RNA-seq on each library, resulting in 227.99 million reads. Then the raw data were preprocessed, and the low-quality data were filtered to remove contamination and joint sequences. The sequenced Clean Data after filtration reached 226.42 million (Q30 > 94.90%). This indicated that RNA-seq has preferable quantity and quality, and this is conducive to comparative analysis with the reference genome of *L. monocytogenes*. Finally, the ratio of reads mapping to *L. monocytogenes* reference genome ranged from 96.54% to 99.61%, showing a good mapping rate (Table 1).

### 3.3. Analysis of DEGs

According to the specifically mapped reads, FPKM values were calculated for the evaluation of relative gene expression levels in *L. monocytogenes* as suspended aggregates and floating cells. DESeq2 software was utilized to explore the difference of gene expression, and the DEGs were screened based on the condition (Corrected significance value of statistical difference: padj < 0.05, expression difference multiple: |log_2_FoldChange| > 1), (Appendix A Appendix A) [27]. Thereafter, some genes were judged to be DEGs in 10%-RIS-treated samples by taking floating cells as the control. Taking group X as the control, we see that there were some significant DEGs in group Y, including 210 upregulated genes and 240 downregulated genes (Appendix A Appendix A).

### 3.4. GO Enrichment Analysis of DEGs

GO includes three ontologies, which describe the cellular component (CC), biological process (BP) and molecular function (MF) of the gene, respectively. GO functional significance enrichment analysis showed which biological functions were significantly associated with DEGs; 30 GO terms with the most obvious enrichment of DEGs were selected. There were five types of CC: ribosome (GO: 0005840), chromosome (GO: 0005694), cell (GO:0005623), DNA topoisomerase complex (ATP-hydrolyzing) (GO: 0009330) and nucleoid (GO: 0009295). Moreover, there were 15 types of BP, the top two of which were translation (GO: 0006412) and glycolytic process (GO: 0006096), of which chemotaxis and bacterial-type flagellum-dependent cell motility terms indicate that suspended aggregate formation was closely related to bacterial chemotaxis and flagellar assembly. There were 10 types of MF; the top two terms were DNA binding (GO: 0003677) and structural constituent of ribosome (GO: 0003735) (Figure 3).

### 3.5. KEGG Enrichment Analysis of DEGs

Figure 4 showed the 30 most significantly enriched (the lowest Qvalue) pathways. The results of KEGG enrichment indicated that there were 19 M (metabolic) pathways, four HD (human disease) pathways, four GIP (genetic information processing) pathways, two OS (organismal system) pathways and one CP (cellular process) pathway. Four of the top five pathways were carbon metabolism (26 DEGs), pyrimidine metabolism (16 DEGs), purine metabolism (15 DEGs) and pyruvate metabolism (13 DEGs), indicating that 10%-RIS treatment stimulated the metabolism of *L. monocytogenes*. This deeply demonstrates that *L. monocytogenes* may change greatly after forming suspended aggregates. Therefore, deeper studies are required to reveal the mechanisms associated with metabolic changes. Ribosome-enriched multiple upregulated genes were in accordance with the results of GO annotation, indicating that the ribosome synthesis pathway was stimulated. In addition, bacterial chemotaxis, flagellar assembly and the two-component system are associated with bacterial motility and have been reported to be associated with bacterial biofilm formation. Further analysis of their relationship with suspended aggregate formation is required.

### 3.6. Analysis of Essential DEGs

Suspended aggregates are regarded as non-classical biofilms. Therefore, a part of gene-expression products may develop an essential impact in the formation of suspended aggregates. According to the bacterial chemotaxis, flagellar assembly and two-component system pathways, 12 DEGs associated with suspended aggregate formation were identified with the cutoff of | log_2_FoldChange | >1 and padj <0.05 (Table 2). Seven of them play roles in the flagellar assembly pathway. Four of them play roles in the bacterial chemotaxis pathway. Nine of them play roles in the bacterial chemotaxis pathway. Some genes play a role in more than one pathway, and these genes also play a pivotal role in each pathway. As shown in Table 2, *cheA* and *cheY* play roles in the bacterial chemotaxis and two-component system with high upregulated expression fold change of nearly 4 and 13, respectively. Based on the gene sequence searched by NCBI, the distance between the two genes was only 20 bp, which was suitable for double gene knockout. Therefore, both genes were knocked out at the same time to verify their roles in the formation of *L. monocytogenes* suspended aggregates.

### 3.7. The Role of cheA and cheY in Strain Growth and Motility of L. monocytogenes

The gene-deletion strain was constructed by using the homologous recombination technique and is referred to as *L. monocytogenes* Δ*cheAY*. There was no significant difference in growth rates at 37 °C between *L. monocytogenes* wild type and *L. monocytogenes* Δ*cheAY* (Figure 5a). It showed that the absence of *cheA* and *cheY* would not affect the growth of *L. monocytogenes*. Then the motility based on chemotaxis was evaluated by quantifying the spread capacity of the strain on semisolid agar plates [28]. The colony diameters of the wild-type and mutant strains on semisolid agar plates were about 20.9 ± 1.3 mm and 9.5 ± 0.5 mm, respectively, the motility assay could reveal there were significant difference (Figure 5b). These data indicate that the deletion of *cheA* and *cheY* significantly affected the motility of *L. monocytogenes*.

### 3.8. cheA and cheY are Involved in Suspended Aggregate Formation in L. monocytogenes

Considering the essential role of *cheA* and *cheY* in bacterial motility, we investigated the effect of *cheA* and *cheY* in suspended aggregate formation in the studied *L. monocytogenes* strains. With the promotion of 10% RIS, *L. monocytogenes* Δ*cheAY* could hardly form suspended aggregates compared with the wild strain, and the aggregation indices decreased from 52.92 ± 2.29% to 0. As a result, compared with *L. monocytogenes*, the ability of suspended aggregate formation of *L. monocytogenes* Δ*cheAY* was significantly reduced, even completely lost (Figure 5c). These findings support the conclusion that *cheA* and *cheY* play decisive roles in the formation of *L. monocytogenes* suspended aggregates.

## 4. Discussion

In our previous research, we found that *R. insidiosa* induced suspended aggregate formation of *L. monocytogenes* under the condition of poor nutrition at low speed [16]. Then 10% RIS was used instead of *R. insidiosa* to carry out the experiment. Moreover, the formation of suspended aggregates was also observed. Suspended aggregates formed by *L. monocytogenes* with the presence of 10% RIS was successfully characterized by aggregation indices and scanning electron microscope. This suggests that *R. insidiosa* may secrete one or more component in nutrient-poor environments that are key components in inducing *L. monocytogenes* to form suspended aggregates. In the actual environment, *R. insidiosa* can continuously produce key components, which continuously induce *L. monocytogenes* to form suspended aggregates, enhancing the resistance of *L. monocytogenes* to environmental conditions.

To dissect the gene related to the formation mechanism of suspended aggregates, we analyzed the transcriptome data from *L. monocytogenes* grown in 10% TSB medium with the presence of 10% RIS. Starting from 6 sequenced RNA libraries, we obtained 240 upregulated DEGs and 210 downregulated DEGs. Based on the DEGs obtained, GO- and KEGG-enrichment analyses were carried out to investigate the effect of 10% RIS on the formation of *L. monocytogenes* suspended aggregates and the potential molecular mechanism. These genes were mainly related to the two-component system, bacterial chemotaxis and flagellar assembly, which had an intense response to 10% RIS treatment. The GO-function-significance-enrichment analysis provided the GO function items that were significantly enriched in DEGs compared with the genome of planktonic cells, so as to give the biological functions that were significantly correlated with DEGs in *L. monocytogenes* under the effect of 10% RIS. According to the results of GO-enrichment analysis, it can be concluded that the presence of 10% RIS affects the metabolic function of *L. monocytogenes* in suspended aggregates, mainly reflected in glycolysis process and amino acid transport. Through the significant enrichment of the KEGG pathway, we identified 102 biochemical metabolic pathways involved in DEGs. In the study of Fan et al., some gens related to bacterial chemotaxis and flagellar assembly were increased in biofilm cells, compared with planktonic cells [29]. Based on the current literature on biofilm formation [30,31], we mainly studied three of the 102 enriched pathways related to biofilm formation, namely bacterial chemotaxis, flagellar assembly and the two-component system.

The two-component system (TCS; ko02020) is a signal transduction system in bacteria that is involved in the regulation of most physiological processes of bacteria, including environmental adaptation, biofilm formation, bacterial chemotaxis, quorum sensing and expression of virulence genes [32]. Histidine kinase (HK) and response regulator (RR) are the main components of the TCS. They realize signal transduction through the transmission of phosphate groups. Most of HKs are transmembrane proteins, which can sense different environmental signals and are responsible for transferring extracellular signals into the cell. RR is located in the cytoplasm and is the region where phosphorylation occurs. The signal receptor domain of HK receives the signal from the external environment and phosphorylates, and then the phosphoric group is transferred to the aspartic acid residue of the regulatory domain of RR, so as to regulate the expression of downstream genes and generate the corresponding regulatory response [32]. In TCS, CheY is a response regulator consisting of a CheY-like receiver domain and a HTH DNA-binding domain DNA-binding domain, and CheA is two-component sensor histidine kinase. The autophosphorylating activity of CheA may increase by the presence of 10% RIS; then CheY was activated.

Chemotaxis is a chemotactic response produced by organisms to the stimulation of chemical substances in the external environment. The pathway is composed of histidine kinase (CheA), response modifier (CheY), methyltransferase (CheR), methyl esterase (CheB), adaptor protein (CheW) and some chemical receptors (MotA, FliM, etc.) [33]. CheA is a central processing unit in chemotactic pathway, sensing environmental signals through transmembrane chemoreceptors and then autophosphorylation. CheY then receives the phosphate group from CheA and binds to flagellar motor switching proteins, including FliM, FliN and FliG. This causes the flagellar to flip clockwise, causing the bacteria to move in a different direction [34]. According to the KEGG-enrichment pathway of DEGs, bacterial chemotaxis (ko02030) was significantly enhanced, and enriched with four upregulated DEGs. Similarly, cheA/cheY, which also plays a role in TCS of bacterial adaptation to selective pressure [35], was also upregulated by about 4-fold change and 13-fold change, respectively. Moreover, *motA* (8.42-fold change) and *fliM* (15.81-fold change) also showed high upregulated expression in the presence of 10% RIS.

Flagella is a special structure that widely exists on the body surface of bacteria as the result of long-term adaptation in the process of bacterial evolution. Flagellar assembly (ko02040) enables bacteria to better adapt to the environment for their own survival. As shown in Appendix A Appendix A, expression levels of genes related to flagella of *L. monocytogenes* in suspended aggregates increased compared with *L*. *monocytogenes* in the floating state. With the inducement of 10% RIS, the expression of *fliM* gene encoding the bacterial flagellar motion-switching protein was upregulated by 15.81-fold. In addition, the upregulated expression of *fliC* gene (2.45-fold change) was also found. FliC is a flagellar filament structural protein, which is considered to be key to motility of strains and biofilm formation [36]. MotA (8.42-fold change) and MotB proteins play the role of proton transmembrane transport in flagellar motion complex and provide the energy required for flagellar rotation. Together, they form the stator of flagella and interact with FliM, FliG and FliN proteins [37]. Other genes that are upregulated in flagellar assembly include *flhA* (encoding flagellar biosynthetic proteins), *fliR* (encoding flagellar type III secretion system proteins), *flhA* (encoding flagellar biosynthetic proteins), *flgK* (encoding hook-related proteins) and *fliE* (encoding flagellar hook–basal body complex protein).

Based on the results discussed above, the regulatory network of suspended aggregates formed by *L. monocytogenes* with the presence of 10% RIS was summarized in nutrient-poor environments (Figure 6). *L. monocytogenes* was stimulated by one or more component in 10% RIS when it is in nutrient-poor liquid environments. Then histidine kinase CheA, which acts on both TCS and bacterial chemotaxis, was activated and autophosphorylated. CheY then received the phosphate group from CheA to form phospho-CheY, which acts on TCS and bacterial chemotaxis pathway. Phospho-CheY binds to proteins in the flagellar motor switch complex, which consists of FliM, FliN and FliG, resulting in increased expression of flagellar assembly related genes (*fliM*, *motA*, *fliC*, etc.). The rotation direction of flagella is affected, and the original clockwise direction is replaced by counterclockwise direction. The clockwise rotation of the flagellum makes the cell rotate in place and remain stationary, while the counterclockwise rotation makes the flagellum form a single rotating bundle and produce smooth swimming behavior [38]. Under the condition of low-speed culture, the change of rotation direction makes *L. monocytogenes* physically move in a certain direction and may eventually result in the formation of suspended aggregates.

To explore the possible mechanism of *L. monocytogenes* forming suspended aggregates with the presence of 10% RIS by RNA-seq technology, it is also necessary to remove out vital genes related to the suspended aggregate formation to prove the function of vital genes. From the previous analysis, it can be concluded that *cheA* and *cheY* are the vital genes for the formation of suspended aggregates. The distance between *cheA* and *cheY* fragments is only 20 bp. Therefore, based on homologous recombination technology, we knocked out these two genes at the same time to verify their effect on suspended aggregates.

Both *cheA* and *cheY* can directly play roles in TCS and bacterial chemotaxis pathway, which are situated in the bicistronic unit [39]. The insertion inactivation experiment of *cheYA* operon suggests that the colony aggregation ability of *L. monocytogenes* decreased [40]. In our study, the growth of *L. monocytogenes* Δ*cheAY* in TSB medium was not affected, but the movement on 0.3% agar plates was restricted, indicating that the proteins encoded by *cheA* and *cheY* genes play a decisive role in chemotaxis. Moreover, Dons et al. also obtained the same experimental results [30]. In nutrient-poor liquid environments, *L. monocytogenes* Δ*cheAY* can hardly be induced to form suspended aggregates by 10% RIS. The contribution of *cheA* and *cheY* to the suspended aggregates of *L. monocytogenes* is consistent with the study of Liu et. al, which showed that the simultaneous deletion of *cheA* and *cheY* reduced the formation of *Azorhizobium caulinodans* biofilms and the production of EPS [30]. The results of the determination of the ability to form suspended aggregates indicate that *cheA* and *cheY* are the key genes regulating the formation of suspended aggregates, and explain the dependence of the aggregation of *L. monocytogenes* on chemotaxis. It has been documented that bacterial chemotaxis is the tendency to move to environments with higher concentrations of beneficial chemicals or lower concentrations of toxic chemicals [41]. Based on the above experimental results and the literature basis, the preliminary mechanism of the formation of *L. monocytogenes* suspended aggregates may be as following: under the action of 10% RIS with high concentration of specific chemical substances, the loss of chemotaxis genes makes *L. monocytogenes* lose chemotaxis. The scattered-chemotaxis mutants were unable to incline to the key chemicals in 10% RIS, thus failing to form suspended aggregates of *L. monocytogenes*.

## 5. Conclusions

In summary, the previous paper indicates that *L. monocytogenes* cannot form suspended aggregates when incubated alone [16]. However, under the action of 10% RIS, *L. monocytogenes* can form suspended aggregates in low-velocity liquid environments with nutrient deficiency. RNA-seq analysis showed that the expression of genes related to metabolism and motility of *L*. *monocytogenes* in suspended aggregates was significantly different from that in the float state. Suspended aggregates are considered to be non-classical biofilms, and the regulation of suspended aggregates by biofilm-related pathways, including two component systems, bacterial chemotaxis and flagellar assembly, was further analyzed. In addition, according to the sequencing results, some DEGs (*cheA*, *cheY*, *motA*, *fliM*, etc.) may be bound up with the formation of suspended aggregates. The next experiment was to construct a mutant strain of *L. monocytogenes* with deletion of both *cheA* and *cheY*. It was found that these two genes were key genes for suspended aggregate formation, and their deletion prevented *L. monocytogenes* from being induced to form suspended aggregates by 10% RIS. The certain mechanism of *cheA* and *cheY* involved in the formation of suspended aggregates remains to be studied. Reverse transcription–quantitative real-time PCR can be utilized to verify the expression of other genes related to these two genes. In addition, we can also explore the function of other key genes on the formation of suspended aggregates.

## Figures and Tables

**Figure 1 microorganisms-09-02591-f001:**
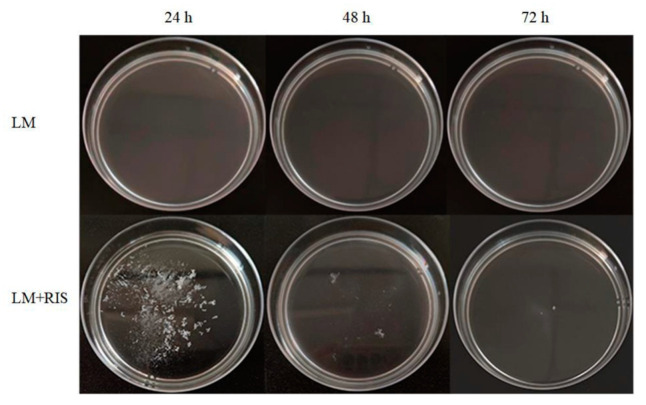
Suspended aggregates formed by *L. monocytogenes* alone or *L. monocytogenes* with the presence of *Ralstonia insidiosa* cell-free supernatants cultured in 10% TSB medium (10% RIS) in 10% TSB at 28 °C for 72 h.

**Figure 2 microorganisms-09-02591-f002:**
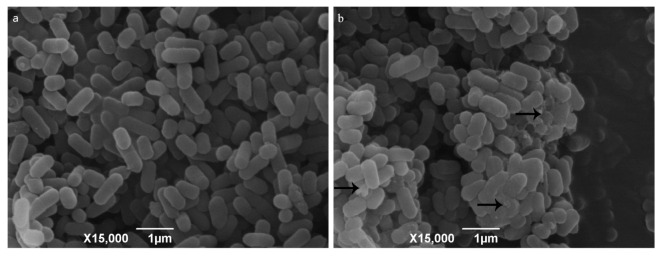
Scanning electron microscope images of 10% RIS treatment. Note: (**a**) *L. monocytogenes* in 10% TSB; (**b**) *L. monocytogenes* with the presence of 10% RIS in 10% TSB. Arrows point to presence of EPS produced by *L. monocytogenes*.

**Figure 3 microorganisms-09-02591-f003:**
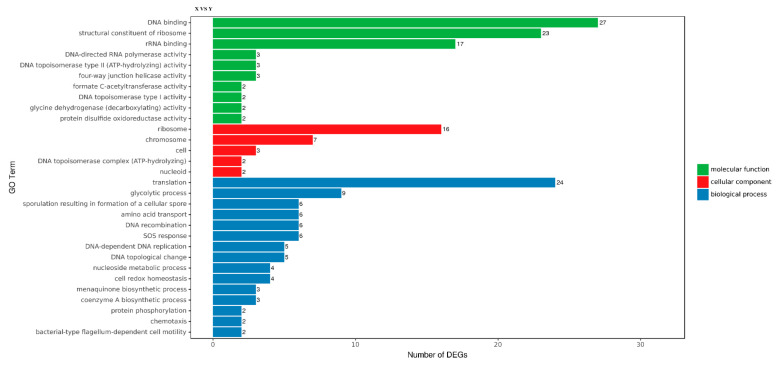
Gene Ontology (GO) distribution map of differentially expressed genes (DEGs) in three main categories. The *y*-axis was the GO term, and the *x*-axis was the number of DEGs in the term. Different colors were utilized to distinguish biological processes, cellular components and molecular functions.

**Figure 4 microorganisms-09-02591-f004:**
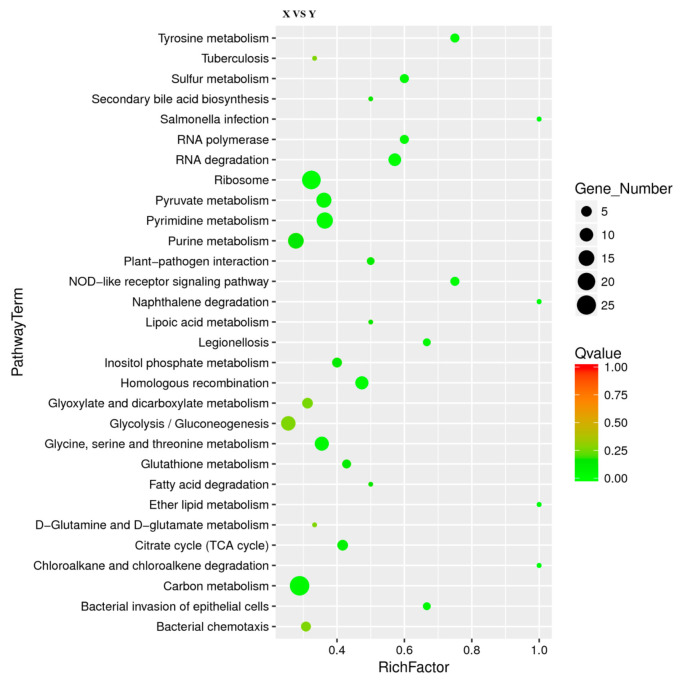
Scatter plot of Kyoto Encyclopedia of Genes and Genomes (KEGG) pathway enrichment for DEGs. The number of DEGs in each pathway is closely related to the size of dots, and the color of dots reflects different Qvalue values. Rich factor was positively correlated with enrichment degree. The smaller the Qvalue, the more significant the enrichment.

**Figure 5 microorganisms-09-02591-f005:**
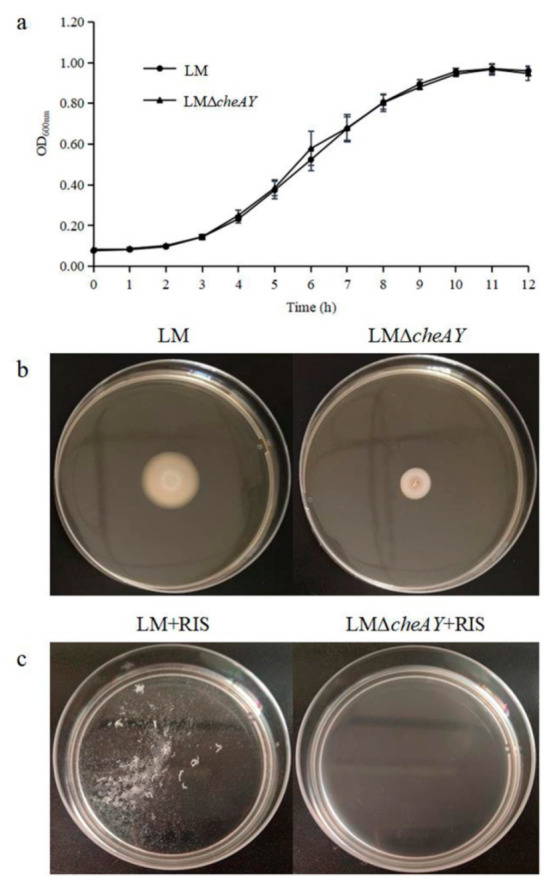
(**a**) Growth curves of *L. monocytogenes* and *L. monocytogenes* Δ*cheAY* at 37 °C. (**b**) Motility of *L. monocytogenes* and *L. monocytogenes*
*ΔcheAY* in semisolid agar. *L. monocytogenes* and *L. monocytogenes* Δ*cheAY* were stabbed into semisolid agar plates (TSB plus 0.3% agar). The plates were cultured at 28 °C for 48 h. (**c**) Suspended aggregates formed by *L. monocytogenes* and *L. monocytogenes* Δ*cheAY* with the presence of 10% RIS in 10% TSB at 28 °C for 24 h.

**Figure 6 microorganisms-09-02591-f006:**
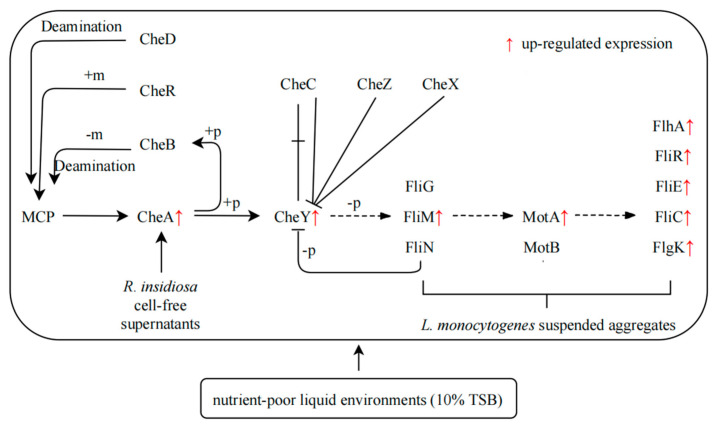
Formation of suspended aggregates of *L. monocytogenes* induced by 10% RIS Scheme 10. TSB environments. +p: Phosphorylation, phosphorylation of a molecule is the attachment of a phosphoryl group. Note: −p, dephosphorylation—as the reverse process of phosphorylation, dephosphorylation is the removal of a phosphate (PO_4_³^−^) group formed by the hydrolysis of organic compounds; +m, methylation—methylation denotes the addition of a methyl group on a substrate, or the substitution of an atom (or group) with a methyl group; −m, demethylation—demethylation is the chemical process of removing methyl group (CH_3_) from the molecule; deamination—deamination is the removal of an amino group from a molecule. Enzymes that catalyze this reaction are called deaminases.

**Table 1 microorganisms-09-02591-t001:** Data-quality statistics and reference sequence matching after filtering.

Sample	Total Reads	Bases	Q20 ^1^ (%)	Q30 ^2^ (%)	GC ^3^ (%)	Total Mapped
X1	36,894,698	5,449,467,565	98.08	94.94	40.97	36,593,927 (99.1848%)
X2	34,176,440	5,052,294,588	98.08	94.90	41.42	32,993,382 (96.5384%)
X3	50,401,832	7,445,052,570	98.08	94.90	41.45	49,339,378 (97.8920%)
Y1	34,084,702	5,025,917,805	98.39	95.66	40.64	33,952,917 (99.6134%)
Y2	34,081,134	5,024,626,651	98.46	95.81	40.79	33,850,013 (99.3219%)
Y3	36,777,634	5,430,302,095	98.41	95.72	40.84	36,502,258 (99.2512%)

^1^ Proportion of the total number of bases with the Phred value greater than 20. ^2^ Proportion of the total number of bases with the Phred value greater than 30. ^3^ Proportion of bases G and C in the total number of bases. X1-3: Non-aggregated *L*. *monocytogenes* samples. Y1-3: Suspended aggregate samples of *L*. *monocytogenes.*

**Table 2 microorganisms-09-02591-t002:** Essential DEGs.

Functional Category	Gene Name	KO_Name	Protein	log_2_FoldChange
Flagellar assembly; bacterial chemotaxis	DR89_RS14910	*fliM*	Flagellar motor switch protein FliM	3.982754445
Flagellar assembly; two-component system	DR89_RS14955	*fliC*	Flagellin	1.294521613
Flagellar assembly	DR89_RS14880	*flgK*	Flagellar hook-associated protein FlgK	3.199252693
Flagellar assembly	DR89_RS14845	*fliE*	Flagellar hook–basal body complex protein FliE	3.692945001
Flagellar assembly	DR89_RS15005	*flhA*	Flagellar biosynthesis protein FlhA	3.37719265
Flagellar assembly	DR89_RS15015	*fliR*	Flagellar type III secretion system protein FliR	3.346243552
Flagellar assembly; bacterial chemotaxis; two-component system	DR89_RS14980	*motA*	Flagellar motor stator protein MotA	3.073232587
Bacterial chemotaxis; two-component system	DR89_RS14945	*cheA*	Chemotaxis family; sensor histidine kinase CheA	1.975793359
Bacterial chemotaxis: two-component system	DR89_RS14950	*cheY*	Chemotaxis family, chemotaxis protein CheY	3.748414246
Two-component system	DR89_RS13310	*liaS*	NarL family, sensor histidine kinase LiaS	2.473963827
Two-component system	DR89_RS03800	*maeA*	NAD-dependent malic enzyme	1.511261314
Two-component system	DR89_RS07650	*cydB*	Cytochrome d ubiquinol oxidase subunit II	−2.073407712
Two-component system	DR89_RS16575	*frdA*	Flavocytochrome c	−2.919069109
Two-component system	DR89_RS08575	*agrA*	DNA-binding response regulator	−1.516198497

## Data Availability

Not applicable.

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
