# Peer review of "Gene Analysis of Listeria monocytogenes Suspended Aggregates Induced by Ralstonia insidiosa Cell-Free Supernatants under Nutrient-Poor Environments"

_microorganisms, 2021, doi:10.3390/microorganisms9122591_

Round 1

Reviewer 1 Report

Li et al.: Gene analysis of Listeria monocytogenes suspended aggregates induced by Ralstonia insidiosa cell-free supernatants under nutrient-poor environments

In the present study, the aggregation of Listeria monocytogenes induced by Ralstonia insidiosa culture supernatants was analyzed at a molecular level. The authors used RNAseq to characterize transcriptomic changes in L. monocytogenes and identified, among others, a strong upregulation of cheA and cheY when aggregation was induced by R. insidiosa supernatant. A gene deletion approach was utilized to demonstrate that both genes together are required for induced aggregation. This research provides new insight on L. monocytogenes cell aggregation which is relevant for applied aspects of food safety.

Some points should be addressed to improve clarity/presentation:

“…aggregates induced by Ralstonia insidiosa cell-free supernatants under nutrient-poor environments” Was it tested whether aggregation takes place in other media than 10% TSB medium?

Line 76: „…. may be cause….“ should read “…. may be caused …..”

Line 82f: “Then, RNA-seq was utilized for preliminarily analyzing the impact of the whole transcriptome of L. monocytogenes forming suspended aggregates induced by 10% RIS.” Clarification required: Analyzing the impact of whole transcriptome on what? Or is the impact of cell aggregation on the transcriptome to be analyzed?

Results section 3.1: Here the authors use cell free supernatant of R. insidiosa to induce aggregation of L. monocytogenes. Was this the first demonstration that aggregation can be induced by cell free supernatant? If yes, it could be made clearer that this is a new observation.

Line 267: “samples were collected under both conditions after the strain formed suspended aggregates in Plastic Petri dishes.” I assume that “both conditions” refers to aggregated and non-aggregated states, of L. monocytogenes, please clarify

Table 1: It appears that samples X1-3 and Y1-3 refer to technical or biological replicates for RNAseq library construction and possibly aggregated and non-aggregated L. monocytogenes. Please clarify

Line 292-296. I believe this is commonly known in the audience of the journal and doesn’t need to be explained in depth

Section 3.7: cheA and cheY were deleted in L. monocytogenes and effects on aggregation analyzed. It should be considered to include a figure showing the operon structure of cheA and cheY in L. monocytogenes. Since both genes were deleted simultaneously, the potential impact of either gene alone on aggregation should be discussed or analyzed. Currently, it is formally shown that both genes together are important for aggregtation but the individual contribution of chaA and cheY remains unknown. 

Since the authors have shown that cheA and cheY are required for the more artificial aggregation induced by cell free supernatant of Ralstonia insidiosa, can it be demonstrated that aggregation induced by whole R. insidiosa cells is also dependent on cheA and cheY?

Line 446: “As shown in Figure S2, expression levels of genes related to flagella of L. monocytogenes in suspended aggregates increased in nutrient-poor environments.” Was expression analyzed in nutrient-poor and nutrient rich environments? If not, this should be expressed differently. It appears that DEG were studied in aggregated and non-aggregated L. monocytogenes but both in nutrient poor environment (10% TSB).

Figure 6: The legend for this figure should be extended to cover all aspects shown. What does +p and -p or +m and -m refer to? “Deamination” should be explained.

Section 5 (Conclusions): Lines 524-528 are not suitable for a conclusion as they rather provide an outlook and suggestion for additional experiments.

Line 512: “In summary, the results of this paper indicate that L. monocytogenes cannot form suspended aggregates when incubated alone.” Is this indeed a key result of the current study? It appears that this is already a conclusion of the previous paper (reference 16)

Line 515: “RNA-seq showed that the metabolism and motility of L. monocytogenes changed largely” This should be expressed differently or removed. RNAseq does not provide direct information about either metabolism or motiliy.

Author Response

Reviewer #1: In the present study, the aggregation of Listeria monocytogenes induced by Ralstonia insidiosa culture supernatants was analyzed at a molecular level. The authors used RNAseq to characterize transcriptomic changes in L. monocytogenes and identified, among others, a strong upregulation of cheA and cheY when aggregation was induced by R. insidiosa supernatant. A gene deletion approach was utilized to demonstrate that both genes together are required for induced aggregation. This research provides new insight on L. monocytogenes cell aggregation which is relevant for applied aspects of food safety.

Response: Thank you for your arduous work and instructive advice.

Specific comments:

Point 1: “…aggregates induced by Ralstonia insidiosa cell-free supernatants under nutrient-poor environments” Was it tested whether aggregation takes place in other media than 10% TSB medium?

Response 1: Thank you for your comments. Yes, we have studied the induction effect of R. insidiosa cell-free supernatants cultured in TSB medium, 20% TSB medium and 10% TSB medium on the formation of L. monocytogenes suspended aggregates. And we found that L. monocytogenes did not form suspended aggregates under the condition of R. insidiosa cell-free supernatants cultured in TSB medium. L. monocytogenes formed suspended aggregates with the presence of R. insidiosa cell-free supernatants cultured in 20%TSB medium and 10% TSB medium, and the aggregation score was the highest in 10% TSB medium. That is, R. insidiosa cell-free supernatants cultured in 10% TSB medium has a strong ability to induce L. monocytogenes to form suspended aggregates.Therefore, combined with the experimental conditions in reference 16, 10% TSB medium was used as the experimental condition in this paper.

Point 2: Line 76: „…. may be cause….“ should read “…. may be caused …..”

Response 2: It was our careless and we have revised the word to “caused”.

Point 3: Line 82: “Then, RNA-seq was utilized for preliminarily analyzing the impact of the whole transcriptome of L. monocytogenes forming suspended aggregates induced by 10% RIS.” Clarification required: Analyzing the impact of whole transcriptome on what? Or is the impact of cell aggregation on the transcriptome to be analyzed?

Response 3: It is the impact of cell aggregation on the transcriptome to be analyzed. And we have revised the sentence to “ Then, RNA-seq was utilized for preliminarily analyzing the impact of the formation of L. monocytogenes suspended aggregates induced by 10% RIS on the transcriptome of L. monocytogenes.”.

Point 4: Results section 3.1: Here the authors use cell free supernatant of R. insidiosa to induce aggregation of L. monocytogenes. Was this the first demonstration that aggregation can be induced by cell free supernatant? If yes, it could be made clearer that this is a new observation.

Response 4: Yes, it was the first demonstration that aggregation can be induced by cell free supernatantant. We have added the sentence named “This is a new observation and the first demonstration that aggregation can be induced by cell-free supernatants.” in Results section 3.1.

Point 5: Line 267: “samples were collected under both conditions after the strain formed suspended aggregates in Plastic Petri dishes.” I assume that “both conditions” refers to aggregated and non-aggregated states, of L. monocytogenes, please clarify

Response 5: Thank you for your advice. We have revised the sentence to “samples were collected under both conditions after the strain formed suspended aggregates or showed floating state in Plastic Petri dishes.”.

Point 6: Table 1: It appears that samples X1-3 and Y1-3 refer to technical or biological replicates for RNAseq library construction and possibly aggregated and non-aggregated L. monocytogenes. Please clarify

Response 6 : It was a valuable advice. We have noted the sentence named “X1-3: Non-aggregated Listeria monocytogenes samples; Y1-3: Suspended aggregate samples of L. monocytogenes.”.

Point 7: Line 292-296. I believe this is commonly known in the audience of the journal and doesn’t need to be explained in depth

Response 7: Thank you for your comments. We have reduced the content of this part appropriately.

Point 8: Section 3.7: cheA and cheY were deleted in L. monocytogenes and effects on aggregation analyzed. It should be considered to include a figure showing the operon structure of cheA and cheY in L. monocytogenes. Since both genes were deleted simultaneously, the potential impact of either gene alone on aggregation should be discussed or analyzed. Currently, it is formally shown that both genes together are important for aggregation but the individual contribution of cheA and cheY remains unknown. 

Response 8: It was a valuable advice. In this manuscript, the effect of simultaneous deletion of cheA and cheY on the formation of L. monocytogenes suspended aggregates was studied. According to the joint key role of these two genes on aggregation, our laboratory will conduct research on the separate role of cheA and cheY in the later stage, and also focus on other genes related to cheA and cheY operon structures. These experiments may be more conducive to displaying the figure of cheA and cheY operon structures.

Point 9: Since the authors have shown that cheA and cheY are required for the more artificial aggregation induced by cell free supernatant of Ralstonia insidiosa, can it be demonstrated that aggregation induced by whole R. insidiosa cells is also dependent on cheA and cheY?

Response 9: This paper mainly studied the effect of R. insidiosa cell-free supernatants on L. monocytogenes aggregation, and there were no related experiments on L. monocytogenes suspended aggregates induced by R. insidiosa. However, we have investigated the effects of cheA and cheY on the induction of R. insidiosa, and found that the detection of cheA and cheY did not affect the ability of R. insidiosa to induce L. monocytogenes. Comparing the key effects of cheA and cheY on the aggregation induced by R. insidiosa cell-free supernatants, there are more than one component or pathway for R. insidiosa to induce the formation L. monocytogenes suspended aggregates. Further experiments are still under study.

Point 10: Line 446: “As shown in Figure S2, expression levels of genes related to flagella of L. monocytogenes in suspended aggregates increased in nutrient-poor environments.” Was expression analyzed in nutrient-poor and nutrient rich environments? If not, this should be expressed differently. It appears that DEG were studied in aggregated and non-aggregated L. monocytogenes but both in nutrient poor environment (10% TSB).

Response 10: Expression was not analyzed in nutrient-poor and nutrient rich environments. Thank you for your advice. We have revised the sentence to “As shown in Figure S2, expression levels of genes related to flagella of L. monocytogenes in suspended aggregates increased compared with L. monocytogenes in the floating state.”.

Point 11: Figure 6: The legend for this figure should be extended to cover all aspects shown. What does +p and -p or +m and -m refer to? “Deamination” should be explained.

Response 11: It was our careless. We have noted the sentence named “+p: Phosphorylation, phosphorylation of a molecule is the attachment of a phosphoryl group. -p: Dephosphorylation, as the reverse process of phosphorylation, dephosphorylation is the removal of a phosphate (PO4³⁻) group formed by the hydrolysis of organic compounds. +m: Methylation, methylation denotes the addition of a methyl group on a substrate, or the substitution of an atom (or group) with a methyl group. -m: Demethylation, demethylation is the chemical process of removing methyl group (CH3) from the molecule. Deamination: Deamination is the removal of an amino group from a molecule. Enzymes that catalyse this reaction are called deaminases.” in Figure 6.

Point 12: Section 5 (Conclusions): Lines 524-528 are not suitable for a conclusion as they rather provide an outlook and suggestion for additional experiments.

Response 12: Thank you for your comments. This paper mainly studied the effect of Ralstonia insidiosa cell-free supernatants induction on analyzing transcriptome data of L. monocytogenes in suspended aggregates and mining related genes by homologous recombination technology in nutrient-poor liquid environment. Therefore, in the conclusion, we prospected the future research of the key genes cheA and cheY and other genes formed with suspended aggregates, and mentioned reverse transcription-quantitative real-time PCR that can quantify gene expression.

Point 13: Line 512: “In summary, the results of this paper indicate that L. monocytogenes cannot form suspended aggregates when incubated alone.” Is this indeed a key result of the current study? It appears that this is already a conclusion of the previous paper (reference 16)

Response 13: It was our careless and we have revised the sentence to “In summary, the previous paper indicates that L. monocytogenes cannot form suspended aggregates when incubated alone [16].”.

Point 14: Line 515: “RNA-seq showed that the metabolism and motility of L. monocytogenes changed largely” This should be expressed differently or removed. RNAseq does not provide direct information about either metabolism or motiliy.

Response 14: Thank you for your advice. We have revised the sentence to “The result of RNA-seq analysis showed that the metabolism and motility of L. monocytogenes changed largely.”.

Reviewer 2 Report

Overall, the manuscript brings some new information to the scientific community and could be of interest for the readers of the journals. Nevertheless I have some concerns that preclude its publication in the present form.

Please see specific comments:

Introduction:

Lines 78-83 – this part describes the methodology and does not fit the section  introduction in my opinion  

Material and methods:

Line 92 -  Please specify here how many strains you used for the study

Line 92 - ,, L. monocytogenes was 92 obtained from the milk sample, with laboratory tab of 100” – please explain this sentence because I do not understand and add some information about identification tested strains from milk.

Also please provide information about  basic characterization of tested strain L. monocytogenes  eg. serogrup, antibiotics susceptibility

Results:

Line 249 – please change Listeria moncytogenes to L. monocytogenes

In my opinion, the work significantly broadens the knowledge about aggregates L. monocytogenes

I have no more comments on the work and after provide a minor correction, I recommend it for publication

Author Response

Response to Reviewer 2 Comments

Reviewer #2: Overall, the manuscript brings some new information to the scientific community and could be of interest for the readers of the journals. Nevertheless I have some concerns that preclude its publication in the present form.

Response: We deeply appreciate your comments of our manuscript. We have tried our best to revise our manuscript according to the comments.

Specific comments:

Introduction:

Point 1: Lines 78-83 – this part describes the methodology and does not fit the section  introduction in my opinion

Response 1: Thank you for your comments. We have revised the sentence to “Based on the research of Guo et al. [16], we evaluated the effect of R. insidiosa cell-free supernatants cultured in nutrient-poor environments on the formation of suspended aggregates formed by L. monocytogenes. RNA sequencing (RNA-seq) can be applied to determine the transcriptional changes in the interactions between pathogenic bacteria and microbial interspecies [21]. Then, RNA-seq was utilized for preliminarily analyzing the impact of the formation of L. monocytogenes suspended aggregates induced by 10% RIS on the transcriptome of L. monocytogenes.”.

Material and methods:

Point 2: Line 92 -  Please specify here how many strains you used for the study

Response 2: It was a valuable advice. We have added the sentence named “E. coli DH5α (Takara Bio Inc., Otsu, Shiga, Japan) was used as the standard plasmid host for cloning procedures and was grown in lysogeny broth (LB; Hope Bio-Technology, Qingdao, China) broth and agar. Ampicillin (100 μg/mL) (Biofroxx, Einhausen, Germany) or erythromycin (3 μg/mL) (Solarbio, Beijing, China) were added to agar media or broth as required.”.

Point 3: Line 92 - ,, L. monocytogenes was obtained from the milk sample, with laboratory tab of 100” – please explain this sentence because I do not understand and add some information about identification tested strains from milk. Also please provide information about  basic characterization of tested strain L. monocytogenes eg. serogrup, antibiotics susceptibility.

Response 3: The strain was isolated from milk samples in the laboratory and identified as L. monocytogenes. Previous laboratory studies showed that this strain has strong biofilm and suspended aggregate formation ability than the other eight strains isolated from different food samples, so this strain was used as the main experimental strain in this paper. The serotype of the strain was 1/2a. And in our experiments, the strain is sensitive to penicillin, chloramphenicol, ampicillin, erythromycin and kanamycin.

Results:

Point 4: Line 249 – please change Listeria moncytogenes to L. monocytogenes

Response 4: It was our careless and we have revised the word to “L. monocytogenes”.